# Quiet Quitting among Nurses Increases Their Turnover Intention: Evidence from Greece in the Post-COVID-19 Era

**DOI:** 10.3390/healthcare12010079

**Published:** 2023-12-29

**Authors:** Petros Galanis, Ioannis Moisoglou, Maria Malliarou, Ioanna V. Papathanasiou, Aglaia Katsiroumpa, Irene Vraka, Olga Siskou, Olympia Konstantakopoulou, Daphne Kaitelidou

**Affiliations:** 1Clinical Epidemiology Laboratory, Faculty of Nursing, National and Kapodistrian University of Athens, 11527 Athens, Greece; pegalan@nurs.uoa.gr (P.G.); aglaiakat@nurs.uoa.gr (A.K.); 2Faculty of Nursing, University of Thessaly, 41500 Larisa, Greece; iomoysoglou@uth.gr (I.M.); iopapathanasiou@uth.gr (I.V.P.); 3Department of Radiology, P. & A. Kyriakou Children’s Hospital, 11527 Athens, Greece; irenevraka@yahoo.gr; 4Department of Tourism Studies, University of Piraeus, 18534 Piraeus, Greece; olsiskou@nurs.uoa.gr; 5Center for Health Services Management and Evaluation, Faculty of Nursing, National and Kapodistrian University of Athens, 11527 Athens, Greece; olympiak1982@hotmail.com (O.K.); dkaitelid@nurs.uoa.gr (D.K.)

**Keywords:** quiet quitting, turnover intention, nurses, workplace

## Abstract

As turnover intention is a strong determinant of actual turnover behavior, scholars should identify the determinants of turnover intention. In this context, the aim of this study was to assess the effect of quiet quitting on nurses’ turnover intentions. Additionally, this study examined the impact of several demographic and job characteristics on turnover intention. A cross-sectional study with 629 nurses in Greece was conducted. The data were collected in September 2023. Quiet quitting was measured with the “Quiet Quitting” scale. In this study, 60.9% of nurses were considered quiet quitters, while 40.9% experienced high levels of turnover intention. Multivariable regression analysis showed that higher levels of quiet quitting increased turnover intention. Moreover, this study found that turnover intention was higher among females, shift workers, nurses in the private sector, and those who considered their workplace understaffed. Also, clinical experience was associated positively with turnover intention. Since quiet quitting affects turnover intention, organizations, policymakers, and managers should address this issue to improve nurses’ intentions to stay at their jobs.

## 1. Introduction

Providing health care is a demanding and stressful process for health care professionals, especially nurses, who experience high rates of burnout, dissatisfaction, anxiety, and depression [1,2,3]. Over time, the management of health care organizations has failed to ensure a working environment in which nurses can work efficiently to meet the needs of their patients. Understaffing of nursing departments, inadequacy of other resources, overtime work, and lack of support from the organization are the most important organizational factors associated with deterioration in the quality and safety of nursing care and the well-being of nurses [4,5,6,7]. The impact of the pandemic, with mass patient admissions, intubations, and deaths, put even more strain on nurses. They found themselves providing care in an unsupportive working environment, which carries the pathologies of the past [7]. Under these particularly difficult circumstances, a percentage of nurses declared their turnover intention, with the percentage being higher during the COVID-19 pandemic [8]. However, more often than other healthcare workers, many nurses eventually decided to stay on the job and instead reduce their effort and performance by quietly quitting [9].

Quiet quitting is a phenomenon that started to emerge during the pandemic period and is beginning to take on worrying dimensions. As the COVID-19 pandemic created an economic suffocation in the market with the continuous lockdowns, many job vacancies were lost. As a result, it has become very difficult for nurses to leave and change their profession. Before the COVID-19 pandemic, it was common for nurses to leave their profession and pursue another career path. Leaving the profession, driven mainly by organizational factors, was not limited to older workers but also included newly graduated nurses who recently joined the nursing profession [10,11]. They are now choosing to stay in the profession and in their current job positions, but implementing quiet quitting. An employee who chooses quiet quitting reduces their effort, performs only highly necessary tasks, does not propose new ideas and practices, does not stay overtime, and does not come to work early. Their goal is to work only as much as necessary to avoid being fired [12,13]. In the health sector, only two studies have investigated, until today, the prevalence and factors associated with the phenomenon of quiet quitting. In the first study involving 1760 healthcare professionals [9], it was found that the percentage of nurses choosing quiet quitting was 67.4%, the highest among both medical staff and other healthcare professionals. In the second study involving 946 nurses [14], job burnout emerged as a predictor leading to quiet quitting, while job satisfaction was found to have a negative effect on the occurrence of this phenomenon. As nurses in the post-COVID era are experiencing very high rates of burnout and job dissatisfaction, higher even than other health professionals, it is very likely that in the future the percentage of nurses choosing quiet quitting will increase dramatically [15]. However, quiet quitting is not limited to the health sector. A major study in the US by the labor analytics and consulting firm Gallup found that 50% of the US workforce has opted for quiet quitting [16].

The effects of quiet quitting can affect both the individual and the health care organization. By reducing their effort and creativity, nurses hinder their professional development, which is a necessary prerequisite for the development of the organization itself. Nurses are the professional group that incorporates innovative ideas and practices into their daily tasks in order to create new knowledge, develop healthcare policies and practices, improve the quality of care, and advance health information technology. Through these innovative actions, nurses improve the patient experience of care, including quality and satisfaction, improving the health of populations, and reducing the per capita cost of healthcare [17]. The impact of quiet quitting on the organization is also linked to patient care, as a reduction in the performance of nurses can lead to a failure to meet the needs of patients and inadequate patient care. There are already studies showing that nurses do not have enough time to meet the needs of patients during their shift [18], a situation that may be exacerbated by quiet quitting. All this can ultimately lead to prolonged hospitalization, resulting in increased costs and risks to patient safety (e.g., hospital-acquired infections). Therefore, this phenomenon will be of great concern to the management of organizations in the coming period, as quiet quitting will increasingly impact business productivity.

The turnover of nurses has been an extremely important issue over time, affecting both the functioning of organizations and public health. Turnover refers to the nurses’ voluntary move from the primary employment position within the same organization, moving to another healthcare organization, or even resigning from the profession and changing job orientation [19]. In departments where there are high turnover rates, the risk of error increases, and patient safety is negatively affected. Also, the cost of filling the vacancy is high for the organization, ranging from $49,000 to $88,000, and involves the process of recruiting new staff and training them, which reduces concurrent productivity [20,21]. Additionally, at the macro-level of healthcare systems, there are negative effects of nurse turnover. Nurses are the largest group of health care professionals within organizations, and at the same time, there are large shortages of nursing staff worldwide, which are estimated to remain high until 2030 [22]. Therefore, the turnover of nurses exacerbates the already existing difficulty for health systems to ensure adequate nursing staffing for their services, as the percentage of nurses who declare their intention to turnover reaches almost 80% [23,24]. The problem of nursing understaffing and a lack of available nursing staff is long-standing, and its solution requires urgent attention. Solutions proposed to retain nursing staff in the profession include innovation, resource availability, opportunities for education, staff engagement, and leadership development [25]. But as long as the organizational factors that contribute to the burnout and dissatisfaction of nursing staff remain, the choice of nurses for quiet quitting and turnover will increase, ultimately undermining any effort to innovate and keep staff in their positions. In essence, health systems and health care organizations are caught in a vicious cycle of understaffing, the inability to fill vacancies, and reduced performance. As the phenomenon of quiet quitting has relatively recently begun to affect the health sector, its research is in progress, and this is the first study to investigate its relationship with turnover intention. Undoubtedly, there is a need for further studies in other countries to confirm a strong link between quiet quitting and nurses’ turnover intentions. As we are in the post-COVID-19 era and hospital operations are now normalized, the behavior of quiet quitting may ultimately be a symptom of fatigue, burnout, or depression that has significantly affected nursing staff during the COVID-19 pandemic. Research regarding the phenomenon of quiet quitting is still preliminary and very limited, and there is only one study linking burnout to quiet quitting [14]. The need to investigate other factors possibly related to quiet quitting behavior is deemed necessary.

The way a healthcare organization is organized and operated can influence nurses’ intentions to leave their jobs. Inadequate resources, understaffing, leadership style, burnout, and low job satisfaction enhance nurses’ turnover intentions [26,27,28,29,30]. During the COVID-19 pandemic, factors such as fear of virus infection, increased stress, working conditions, and reduced organizational support were found to contribute to an increase in nurses reporting their turnover intention [31], with rates being higher than before the pandemic [8].

Since turnover intention is a strong determinant of actual turnover behavior [32,33], identification of factors that influence nurses’ turnover intention is essential to improving the number of nurses that stay in their jobs. Moreover, scholars have not yet investigated the relationship between quiet quitting and turnover intention in nurses. Thus, the aim of this study was to evaluate the effect of quiet quitting on nurses’ turnover intentions. Additionally, this study examined several demographic and job characteristics as potential determinants of turnover intention.

## 2. Materials and Methods

### 2.1. Study Design

A cross-sectional study was conducted using only a convenience sample due to financial and logistic limitations. In particular, an online form of the study questionnaire was created with Google Forms, published on the authors’ social media accounts (Facebook, Instagram, WhatsApp, and LinkedIn), and an invitation was sent to nurses to participate in this study. In this context, the response rate cannot be estimated. The data were collected in September 2023. Since the study questionnaire was in Greek, only nurses who understood the Greek language could participate in this study. The study population included adult nurses in Greece who had been working for at least one year.

Considering a weak association between quiet quitting and turnover intention (odds ratio = 1.50), the number of predictors (n = 7), the alpha error set at 5%, the power level at 95%, and a two-tailed test, a sample size of 503 nurses was necessary for multivariable logistic regression analysis. Similarly, in the multivariable linear regression analysis, considering the low effect size (f^2^ = 0.03) of the quiet quitting on turnover intention, the sample size included 436 nurses. The seven predictors in this study were gender, clinical experience, shift work, full-time job, job sector, understaffed workplace, and levels of quiet quitting. An increase in the sample size was decided to further decrease the random error of measurements.

### 2.2. Measures

Demographic and job characteristics of nurses included gender, age, clinical experience (years), shift work, full-time job, job sector (private or public), and understaffed workplace (no or yes).

Quiet quitting in the sample was measured with the “Quiet Quitting” Scale (QQS) [34]. The valid Greek version of the QQS was used [9]. In this study, Cronbach’s alpha for the Greek version of the QQS was 0.782, while McDonald’s Omega was 0.785. Moreover, the authors found moderate to high statistically significant correlations between the three factors of the QQS, indicating high internal reliability of the tool. The intraclass correlation coefficients for the tool and the three factors were greater than 0.97 (*p*-value < 0.001 in all cases) in the test–retest study. Additionally, confirmatory factor analysis proved the three-factor structure of the QQS since all goodness-of-fit statistics showed acceptable values. The QQS includes nine items assigned values from 1 to 5 (Table 1). Moreover, the QQS includes three factors: detachment (four items), lack of initiative (three items), and lack of motivation (two items). Higher values are indicative of higher levels of quiet quitting. Additionally, nurses with a value on the QQS higher than 2.06 are considered quiet quitters [35]. In this study, Cronbach’s alpha for the QQS was 0.82, indicating very good internal reliability. Moreover, Cronbach’s alpha for the factor’s detachment, lack of initiative, and lack of motivation were 0.85, 0.79, and 0.81, respectively.

Turnover intention was measured with the question, “How often have you seriously considered leaving your current job?” [36]. This question is a valid tool to measure turnover intention since the authors found high levels of reliability, convergent validity, and discriminant validity. The valid Greek version of the QQS tool was used [37], and the authors found that the tool has excellent reliability (intraclass correlation coefficient = 0.986, *p*-value < 0.001). Moreover, the tool had excellent concurrent validity since they found moderate and statistically significant correlations between the tool and two other scales (the Patient Health Questionnaire-4 and the COVID-19 burnout scale). Answers are rated on a six-point Likert scale: never (1), rarely (2), sometimes (3), somewhat often (4), quite often (5), and extremely often (6). Values ≥4 denote a high level of turnover intention, while values ≤3 denote a low level of turnover intention. Overall turnover intention ranges from 1 to 6, with higher values indicating higher levels of turnover intention.

Since questionnaires were used to collect all study variables, clear instructions were given to nurses to avoid common method bias [38]. In particular, the presentation for each study variable (i.e., demographic, characteristics, job characteristics, Quiet Quitting Scale, turnover intention scale) was clear and comprehensive by using different parts and colors in the study questionnaire. Moreover, items were not complex or ambiguous, keeping the study questionnaire concise. Also, the link between quiet quitting and turnover intention was concealed from the nurses.

### 2.3. Ethical Considerations

The study protocol was approved by the Ethics Committee of the Faculty of Nursing, National and Kapodistrian University of Athens (approval number: 459, 20 September 2023). Moreover, the personal data of the participants were not collected. Additionally, nurses gave their informed consent before their participation. This study was conducted in accordance with the Declaration of Helsinki [39].

### 2.4. Statistical Analysis

Numbers and percentages were used to present categorical variables. Moreover, mean, standard deviation (SD), minimum value, and maximum value were used to present continuous variables. The Kolmogorov-Smirnov test and Q-Q plots were employed to assess the distribution of continuous variables. Age, QQS score, and turnover intention score followed a normal distribution, while clinical experience did not follow a normal distribution. The bivariate relationship between quiet quitting and turnover intention was assessed with a chi-square trend test. Demographic characteristics, job characteristics, and quiet quitting were the independent variables, while turnover intention was the dependent variable. First, a logistic regression analysis was performed, considering turnover intention as a dichotomous variable. Then, a linear regression analysis was performed, considering turnover intention as a continuous variable, to validate the results. In both cases, univariate and multivariable regression analyses were performed. By estimating a final multivariable regression, the independent effect of quiet quitting and other independent variables on turnover intention was estimated. Regarding logistic regression analysis, unadjusted and adjusted odds ratios (ORs), 95% confidence intervals (Cis), and *p*-values were presented. Also, unadjusted and adjusted beta coefficients, 95% confidence intervals, and *p*-values were presented in the case of linear regression analysis. *p*-values less than 0.05 were considered statistically significant. IBM SPSS 21.0 (IBM Corp. Released 2012. IBM SPSS Statistics for Windows, Version 21.0. IBM Corp., Armonk, NY, USA) was used for the analysis.

## 3. Results

### 3.1. Socio-Demographic Characteristics

The study population included 629 nurses. The mean age of the sample was 39.7 years (SD = 9.8), with a range from 22 to 74 years. Most nurses were female (77.3%) with a full-time job (92.5%). The mean clinical experience was 15.4 years (SD = 12.8), with a range from 1 to 40. Among nurses, 37.7% were shift workers, and 59% have been working in the public sector. Three out of four nurses considered their workplace understaffed. The detailed socio-demographic characteristics of the nurses are shown in Table 2.

### 3.2. Quiet Quitting

The mean score on the QQS was 2.3 (SD = 0.6), with a range from 1.0 to 4.6 and a median value of 2.2. In the sample, 60.9% (n = 383) of nurses had a QQS score above the cut-off point of 2.06 and were considered quiet quitters. On the other hand, 39.1% (n = 246) had a QQS score below the suggested cut-off point and were described as non-quiet quitters. Regarding the three factors of the QQS, mean values for detachment, lack of initiative, and lack of motivation were 2.0 (SD = 0.7), 2.3 (SD = 0.8), and 2.8 (SD = 1.0), respectively. Thus, nurses experienced higher levels of a lack of motivation and initiative and lower levels of detachment.

### 3.3. Turnover Intention

The mean turnover intention was 3.3 (SD = 1.6), with a range from 1.0 to 6.0 and a median value of 3.0. Among nurses, 40.9% (n = 257) considered quitting their current job somewhat, quite, or extremely often, implying a high level of turnover intention. However, 59.1% (n = 372) of nurses in the sample never, rarely, or sometimes considered leaving their job, indicating a low level of turnover intention.

Descriptive statistics for the bivariate relationship between quiet quitting and turnover intention are shown in Table 3. Among quiet quitters, 49.9% (n = 191) reported a high level of turnover intention, while the corresponding percentage among non-quiet quitters was 26.8% (n = 66) (chi-square = 63, *p*-value for trend test < 0.001).

### 3.4. Regression Analysis

Multivariable logistic and linear regression analyses confirmed results from the bivariate analysis regarding the relationship between quiet quitting and turnover intention. In particular, multivariable logistic regression analysis identified that a high level of turnover intention was 3.18 times more common among quiet quitters than non-quiet quitters (adjusted odds ratio = 2.69, 95% CI = 1.87 to 3.86, *p*-value < 0.001). Similarly, multivariable linear regression analysis showed that higher levels of quiet quitting increased turnover intention score (adjusted coefficient beta = 0.89, 95% CI = 0.72 to 1.07, *p*-value < 0.001).

Among other socio-demographic characteristics of nurses, this study found that turnover intention was higher among females, shift workers, nurses in the private sector, and those who considered their workplace understaffed. Also, clinical experience was associated positively with turnover intention. Table 4 and Table 5 show the detailed results from the logistic and linear regression analyses, respectively.

## 4. Discussion

This study has highlighted the very high percentage of nurses who have chosen quiet quitting in their work. This finding is consistent with that of a study in Greece after the COVID-19 pandemic, where nurses were also found to choose quiet quitting at a very high rate [9]. As the phenomenon of quiet quitting has recently emerged in the health sector, in the only study that investigated this phenomenon, burnout was found to influence its occurrence, while job satisfaction was found to be an inhibitor [14]. Studies are also limited in the business sector. The main situations identified as predictors of quiet quitting in the business sector include a lack of: feeling cared about, opportunities to learn, employee autonomy, growth, and connection with the organization’s purpose. The role of feeling undervalued and underappreciated is also important [40,41]. All of the above factors are actually included in the conceptual framework of perceived organizational support [42]. The degree of organizational support received by nurses influences their job satisfaction, commitment to the organization, burnout, and turnover intention [43,44,45]. Although, as mentioned earlier, there are no studies investigating the factors leading to quiet quitting, the existing working conditions are conducive to the development of quiet quitting by nurses. As the nurses see that over the years their working environment is not improving and the effects are becoming more and more burdensome, they may have decided to take action on their own. They choose quiet quitting in an attempt at self-preservation and work–life balance. They are translating all the dissatisfaction and burnout from their work into an attitude that will not get them fired, securing their paycheck while returning home with less of a burden [13,46].

According to the statistically significant findings of this study, nurses who chose quiet quitting also reported their turnover intention. Therefore, it is possible that quiet quitting is a temporary solution, a solution of necessity, until nurses find a better working environment. As long as the administrations of health care organizations fail to address organizational weaknesses, nurses will be driven to quiet quitting and will seek to flee at the first job opportunity.

Nursing understaffing, which was found to influence turnover intention according to this study, is one of the most important factors affecting nurses’ well-being. When nurses work in understaffed departments, they are more likely to experience burnout and dissatisfaction and to report a higher intent to leave their job [47]. Working on shift work, which was found in this study to affect turnover intention, is exacerbated by understaffing. The fewer nurses working, the more night and weekend shifts are assigned to existing staff. The night shifts affect nurses’ intentions to leave work [48]. Over the years, as understaffing and shift work persist, the burden accumulates, especially for nurses with more years of experience. The results of the present study showed a positive relationship between clinical experience and turnover intention. This finding is consistent with the findings of other studies [8,26]. Work experience is a protective factor that can reduce nurses’ burnout and job dissatisfaction [49,50]. However, within the extremely difficult working conditions during the COVID-19 pandemic, where nurses often felt unable to cope with the demands of work [51], it seems that work experience was not enough to protect them. The Greek health system, moreover, has for decades been one of the most understaffed (in terms of nursing staff) among developed countries [52]. Consequently, the most experienced nurses are also the most fatigued, and they express their turnover intentions.

This study has several limitations. First, a convenience sample of nurses was used in Greece. Although the minimum sample size requirements were achieved, further studies with representative and stratified samples would add valuable information. Additionally, studies in other countries and cultures can expand the research question. Second, since a cross-sectional study was performed, a causal relationship between quiet quitting and turnover intention cannot be established. Longitudinal studies following a sample of nurses over time can produce more valid results. Third, a single item to measure turnover intention was used. Although this tool is valid, scholars can also use other valid instruments to measure turnover intention. Fourth, self-reported questionnaires to measure quiet quitting and turnover intention were used. Thus, information bias can arise in this study. Measurement of actual turnover behavior in nurses can reduce this information bias. Fifth, the impact of several demographic and job variables on turnover intention was investigated. However, several other factors can affect turnover intention, e.g., job, personality characteristics, wage, and managers’ attitudes. Sixth, this study investigated for the first time the impact of quiet quitting on turnover intention in a sample of nurses. Thus, generalizations of these results should not be applied. Future studies with less bias and different samples are necessary to improve knowledge. Finally, social desirability bias is probable in this study since nurses may answer the questions regarding quiet quitting in a way that will be viewed favorably by others. However, since an anonymous online survey was conducted, social desirability bias is expected to be low. In any case, further studies should be conducted with larger and random samples to further decrease social desirability bias.

### Implications for the Nursing Profession

This study has highlighted two important implications that should mobilize policymakers and health organization administrations. The first relates to the high rates of nurses choosing to reduce their performance through quiet quitting. Already in the health care sector, there are problems with the quality of patient care, where a high number of errors and unmet patient needs are recorded. Nurses, by choosing quiet quitting, give the false impression that the departments are adequately staffed. The choice of quiet quitting places a workload on those who do not choose this behavior. This creates a fragile work environment where conflicts between staff and management can arise. These conflicts can damage team morale, take valuable time from patient care, and degrade the quality of care.

The second concerns the intention of nurses to leave, even those who choose quiet quitting. When nurses with the most clinical experience also choose to leave, the quality of care provided is also affected. The loss of nurses from many already understaffed departments exacerbates this situation, burdens those who remain in their jobs, and wastes the organization’s resources on recruiting new staff. Health care organizations are under constant pressure to reduce costs while increasing efficiency.

Improving nurses’ working environment combined with organizational support can increase nurses’ satisfaction, reduce burnout, and counteract the current situation. Further studies are needed in other countries regarding the prevalence of quiet quitting and the factors contributing to its occurrence.

## 5. Conclusions

Findings in this study highlighted the high proportions of nurses who choose quiet quitting and their turnover intention, and they also revealed a positive impact of quiet quitting on turnover intention. The phenomenon of quiet quitting is spreading and undermining the efficiency and quality of health care services. Nurses’ working conditions seem to favor this behavior. Although quiet quitting is a solution chosen by many nurses, it is not enough to keep them in the profession, and at the first opportunity, they declare their willingness to leave. The pandemic has found health systems with many organizational problems. There is now no room for the management of health organizations to turn a blind eye to the current situation. Organizational support for nurses and securing the required human and other necessary resources are imperative to creating a working environment that will motivate nurses and stop them from leaving the profession.

## Figures and Tables

**Table 1 healthcare-12-00079-t001:** The nine items that were included in the Quiet Quitting Scale.

1. I do the basic or minimum amount of work without going above and beyond.
2. If a colleague can do some of my work, then I let him/her do it.
3. I take as many breaks as I can.
4. How often do you pretend to be working in order to avoid another task?
5. I don’t express opinions and ideas about my work because I am afraid that the manager assigns me more tasks.
6. I don’t express opinions and ideas about my work because I think that work conditions are not going to change.
7. How often do you take initiative at your work?
8. I find motives in my job.
9. I feel inspired when I work.

**Table 2 healthcare-12-00079-t002:** Socio-demographic characteristics of the nurses.

Characteristics	N	%
Gender		
Females	486	77.3
Males	143	22.7
Age ^a^	39.7	9.8
Clinical experience (years) ^a^	15.4	12.8
Shift work		
No	392	62.3
Yes	237	37.7
Full-time job		
No	47	7.5
Yes	582	92.5
Sector		
Private	258	41.0
Public	371	59.0
Understaffed workplace		
No	163	25.9
Yes	466	74.1

^a^ mean and standard deviation.

**Table 3 healthcare-12-00079-t003:** Bivariate relationship between quiet quitting and turnover intention.

Quiet Quitters	How Often Have You Seriously Considered Leaving Your Current Job?					
	Never	Rarely	Sometimes	Somewhat Often	Quite Often	Extremely Often
No	61 (24.8)	63 (25.6)	56 (22.8)	21 (8.5)	19 (7.7)	26 (10.6)
Yes	28 (7.3)	58 (15.1)	106 (27.7)	66 (17.2)	62 (16.2)	63 (16.4)

Values are expressed as n (%).

**Table 4 healthcare-12-00079-t004:** Logistic regression analysis with turnover intention (dichotomous variable) as the dependent variable (reference category: low level of turnover intention).

Independent Variables	Unadjusted OR	95% Confidence Interval for OR	*p*-Value	Adjusted OR ^a^	95% Confidence Interval for OR	*p*-Value
Females vs. males	1.43	1.04 to 1.98	0.030	1.53	1.001 to 2.33	0.049
Clinical experience (years)	1.004	0.99 to 1.02	0.563	1.01	0.99 to 1.03	0.190
Shift work	2.43	1.86 to 3.17	<0.001	1.71	1.15 to 2.55	0.008
Full-time job	0.64	0.36 to 1.15	0.132	1.28	0.63 to 2.62	0.500
Private sector	0.98	0.74 to 1.28	0.857	1.66	1.11 to 2.48	0.013
Understaffed workplace	3.42	2.41 to 4.88	<0.001	3.04	1.92 to 4.82	<0.001
“Quiet Quitting” Scale score	3.18	2.39 to 4.24	<0.001	2.69	1.87 to 3.86	<0.001

OR: odds ratio. ^a^ R^2^ for the multivariable model = 17.0%.

**Table 5 healthcare-12-00079-t005:** Linear regression analysis with turnover intention (continuous variable) as the dependent variable.

Independent Variables	Unadjusted Coefficient Beta	95% Confidence Interval for Beta	*p*-Value	Adjusted Coefficient Beta ^a^	95% Confidence Interval for Beta	*p*-Value
Females vs. males	0.32	0.06 to 0.57	0.014	0.37	0.11 to 0.64	0.007
Clinical experience (years)	0.01	−0.01 to 0.02	0.354	0.01	0.001 to 0.03	0.033
Shift work	0.85	0.65 to 1.05	<0.001	0.42	0.16 to 0.68	0.002
Full-time job	0.26	−0.19 to 0.70	0.253	0.05	−0.38 to 0.49	0.815
Private sector	0.06	−0.16 to 0.28	0.575	0.31	0.06 to 0.56	0.017
Understaffed workplace	1.09	0.86 to 1.33	<0.001	0.79	0.52 το 1.07	<0.001
“Quiet Quitting” Scale score	1.01	0.86 to 1.15	<0.001	0.89	0.72 to 1.07	<0.001

^a^ R^2^ for the multivariable model = 23.1%, *p*-value for ANOVA < 0.001.

## Data Availability

The data presented in this study are available on request from the corresponding author. This manuscript was drafted against the (STROBE) for a cross-sectional study, descriptive research.

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
