# Peer review of "Quiet Quitting among Nurses Increases Their Turnover Intention: Evidence from Greece in the Post-COVID-19 Era"

_healthcare, 2023, doi:10.3390/healthcare12010079_

Round 1

Reviewer 1 Report

Comments and Suggestions for Authors

A paper presenting novel idea and treating turnover intention in a novel manner.

Factors considered in perceived organizational support are not clearly mentioned.

The study lacks sound . scholarly theoretical basis.

Justification from literature regarding a weak link between quit quitting and turnover intentions has not been provided.

Which are the 7 predictors on the basis of which the sample size has been calculated?

Component variables of the quiet quitting scale score have not been discussed.

While the paper is interesting and novel, the above missing links interfere with the acceptance of the results and discussion etc.

Comments on the Quality of English Language

The paper has a few basic grammatical errors(nurses-'which' instead of 'who').While these do not interfere with the meaning of the paper, a native  of english might underrate the quality of the paper on that basis. It is suggested to go over the grammatical part all over.

Author Response

We thank reviewer 1 for the comments and we submit our point-to-point response

Reviewer 2 Report

Comments and Suggestions for Authors

This study examines a timely and very important area of concern among nurses and is likely to be of great interest among readers - both nurses and administrators/employers (Section 4.1 is particularly important in this case, as were the notes about re-staffing costs given in the introduction). It is a logical extension of the authors previous study conducted in June 2023 (#14).

One concern: the content of Tables 3 and 4 are identical, even though one is logistic and one is linear modeling - check the values within each table and adjust as required.

Author Response

We thank reviewer 2 for the comments and we send our point-to-point response

Reviewer 3 Report

Comments and Suggestions for Authors

- One of the major concerns regarding this research is social desirability. The researchers need to convince the reviewers how they managed internal validity of the results and how they controlled biases mainly social desirability. 

 - Power analysis is needed. 

- The authors need to reflect more thoroughly on psychometrics of measures used. 

Author Response

We thank reviewer 3 for the comments and we send our point-to-point response

Reviewer 4 Report

Comments and Suggestions for Authors Dear Authors, Congratulations on your research topic. This is an important issue regarding the quality of the nurse's work, patient safety and professional satisfaction of the nurse. The described phenomenon is multifactorial, as shown in the introduction to the issue. The purpose of the study was presented. The research design has significant limitations, which have been pointed out. In the future, this requires repeating the research using other research tools to make the results reliable. The statistical analysis is correct and the conclusions relate to the results. The literature contains 50 items, of which only 10 items come from a period older than the last 5 years.

Author Response

We thank reviewer 4 for the comments and we send our point-to-point response

Reviewer 5 Report

Comments and Suggestions for Authors

I have reviewed the manuscript titled “Quiet quitting among nurses increases their turnover intention: 2 Evidence from Greece in the post-COVID-19 era”. It is an interesting manuscript written by the authors however, I have come up with the following observations;

1.       While writing avoid using words like I, my, our, etc. see you have written that “. In this context, our aim was to assess…….”. This seems non-academic. Please write like this; “The aim of the study is to …………. See page 1, line 16.

2.       The authors have written that “Multivariable regression analysis showed a positive relationship between…….”. I am unable to understand that how the regression analysis is used to examine the relationships? It must be effect/impact.

3.       In whole of the document, especially the introduction section and the literature section the authors have used a lot of citations/references that are older than the previous five years. It is recommended to use the latest citations/references (with in the past four years).

4.       I feel that developing the arguments for this research on the basis of COVID-19 are not much stronger point to do. COVID 19 has gone several years back and the things are returned to normal now. It is bit confusing that why the nurses are showing the behaviors of quite quitting? Justification I s required.

5.       Moreover, it is not clear that how much percentage of nurses have actually shown this behavior. Please report some percentage if possible. How you came to know that this is actually happening?

6.       See lines 53-55, the authors have written that “The employee who chooses quiet quitting reduces his/her effort, performs the highly necessary tasks, does not propose new ideas and practices, does not stay overtime and does not come to work early…….”. there can be several other causes of adopting such behavior other than COVID-19. Please read the relevant literature and add the meaningful arguments. the reasons for this behavior can be the presence of nurses from other countries due to opening of borders after COVID-19. Availability of technology and other resources that makes work easy and so forth. please I suggest to develop the arguments based on the reasons other than COVID 19.

7.       The authors have not mentioned that what the earlier studies have found about the quite quitting? What reasons they have provided for adoption of quite quitting behaviors among employees especially nurses.

8.       The authors have missed out providing a reason for using convenience sampling. It is therefore required to provide the justification for using convenience sampling.

9.       How authors managed to control for the Common Method Bias, as they have used questionnaires, a single means of data collection, to collect responses.

10.   In the study the authors have not mentioned the items for measuring quote quitting and turnover intentions. This makes it difficult to understand the whole story. It is suggested to provide the items in a table.

11.   Results are fine. However, there is a need to explain how the demographics information is connected to the whole study, especially the examining the effect of Quite quitting and turnover intentions?

12.   Rarely the latest studies are used by the authors, it is recommended to revise the discussion section.

13.   See the conclusion section lines 317, 318, the authors have written that “The phenomenon of quiet quitting is spreading and undermining the efficiency and quality of health care services. Nurses' working conditions seem to favor this behavior……”. The authors have not explained what those working conditions are that actually are responsible for producing “quite quitting” behavior.

14.   Authors have missed out writing about the  limitations and future directions for this study.

15.   Checking and ensuring the allowed limit of plagiarism is the responsibility of the authors.

16.   Major Revisions.

Comments on the Quality of English Language

I have reviewed the manuscript titled “Quiet quitting among nurses increases their turnover intention: 2 Evidence from Greece in the post-COVID-19 era”. It is an interesting manuscript written by the authors however, I have come up with the following observations;

1.       While writing avoid using words like I, my, our, etc. see you have written that “. In this context, our aim was to assess…….”. This seems non-academic. Please write like this; “The aim of the study is to …………. See page 1, line 16.

2.       The authors have written that “Multivariable regression analysis showed a positive relationship between…….”. I am unable to understand that how the regression analysis is used to examine the relationships? It must be effect/impact.

3.       In whole of the document, especially the introduction section and the literature section the authors have used a lot of citations/references that are older than the previous five years. It is recommended to use the latest citations/references (with in the past four years).

4.       I feel that developing the arguments for this research on the basis of COVID-19 are not much stronger point to do. COVID 19 has gone several years back and the things are returned to normal now. It is bit confusing that why the nurses are showing the behaviors of quite quitting? Justification I s required.

5.       Moreover, it is not clear that how much percentage of nurses have actually shown this behavior. Please report some percentage if possible. How you came to know that this is actually happening?

6.       See lines 53-55, the authors have written that “The employee who chooses quiet quitting reduces his/her effort, performs the highly necessary tasks, does not propose new ideas and practices, does not stay overtime and does not come to work early…….”. there can be several other causes of adopting such behavior other than COVID-19. Please read the relevant literature and add the meaningful arguments. the reasons for this behavior can be the presence of nurses from other countries due to opening of borders after COVID-19. Availability of technology and other resources that makes work easy and so forth. please I suggest to develop the arguments based on the reasons other than COVID 19.

7.       The authors have not mentioned that what the earlier studies have found about the quite quitting? What reasons they have provided for adoption of quite quitting behaviors among employees especially nurses.

8.       The authors have missed out providing a reason for using convenience sampling. It is therefore required to provide the justification for using convenience sampling.

9.       How authors managed to control for the Common Method Bias, as they have used questionnaires, a single means of data collection, to collect responses.

10.   In the study the authors have not mentioned the items for measuring quote quitting and turnover intentions. This makes it difficult to understand the whole story. It is suggested to provide the items in a table.

11.   Results are fine. However, there is a need to explain how the demographics information is connected to the whole study, especially the examining the effect of Quite quitting and turnover intentions?

12.   Rarely the latest studies are used by the authors, it is recommended to revise the discussion section.

13.   See the conclusion section lines 317, 318, the authors have written that “The phenomenon of quiet quitting is spreading and undermining the efficiency and quality of health care services. Nurses' working conditions seem to favor this behavior……”. The authors have not explained what those working conditions are that actually are responsible for producing “quite quitting” behavior.

14.   Authors have missed out writing about the  limitations and future directions for this study.

15.   Checking and ensuring the allowed limit of plagiarism is the responsibility of the authors.

16.   Major Revisions.

Author Response

We thank reviewer 5 for the comments and we send our point-to-point response
